# Finding a Husband: Using Explainable AI to Define Male Mosquito Flight Differences

**DOI:** 10.3390/biology12040496

**Published:** 2023-03-24

**Authors:** Yasser M. Qureshi, Vitaly Voloshin, Luca Facchinelli, Philip J. McCall, Olga Chervova, Cathy E. Towers, James A. Covington, David P. Towers

**Affiliations:** 1School of Engineering, University of Warwick, Coventry CV4 7AL, UK; 2Royal Botanical Garden, Kew, London TW9 3AE, UK; 3Liverpool School of Tropical Medicine, Pembroke Place, Liverpool L3 5QA, UK; 4University of College London Cancer Institute, University College London, London WC1E 6DD, UK

**Keywords:** mosquitoes, machine learning, trajectory analysis, explainable artificial intelligence, mosquito behaviour, classification, insect tracking

## Abstract

**Simple Summary:**

Mosquitoes are vectors of some of the world’s deadliest diseases. A wide range of intervention tools are available, but the majority of these depend on insecticides and must be updated frequently in response to the rise of insecticide resistance in the targeted mosquito populations. The behaviour of mosquitoes is an important factor to add to their ongoing understanding as a vector for disease. The method presented within this paper is able to identify key behavioural differences between male, female, and in copula mosquito pairs by analysing their flight tracks. To identify these differences, we developed a framework that extracts features from track segments. Track segments are used to unify durations and each segment is individually classified as either male or non-male by a machine learning model. The segments for each track are combined to return an overall prediction of the class of the track. This approach is one of the first applications of machine learning to mosquito trajectory analysis. The framework can be extended to analyse differences between other classes of trajectories.

**Abstract:**

Mosquito-borne diseases account for around one million deaths annually. There is a constant need for novel intervention mechanisms to mitigate transmission, especially as current insecticidal methods become less effective with the rise of insecticide resistance among mosquito populations. Previously, we used a near infra-red tracking system to describe the behaviour of mosquitoes at a human-occupied bed net, work that eventually led to an entirely novel bed net design. Advancing that approach, here we report on the use of trajectory analysis of a mosquito flight, using machine learning methods. This largely unexplored application has significant potential for providing useful insights into the behaviour of mosquitoes and other insects. In this work, a novel methodology applies anomaly detection to distinguish male mosquito tracks from females and couples. The proposed pipeline uses new feature engineering techniques and splits each track into segments such that detailed flight behaviour differences influence the classifier rather than the experimental constraints such as the field of view of the tracking system. Each segment is individually classified and the outcomes are combined to classify whole tracks. By interpreting the model using SHAP values, the features of flight that contribute to the differences between sexes are found and are explained by expert opinion. This methodology was tested using 3D tracks generated from mosquito mating swarms in the field and obtained a balanced accuracy of 64.5% and an ROC AUC score of 68.4%. Such a system can be used in a wide variety of trajectory domains to detect and analyse the behaviours of different classes, e.g., sex, strain, and species. The results of this study can support genetic mosquito control interventions for which mating represents a key event for their success.

## 1. Introduction

Over 80% of the global population lives in areas that are at risk from at least one major vector-borne disease [1], with vector-borne diseases accounting for more than 17% of all human infectious diseases [2]. Globally, mosquito-borne diseases such as dengue are increasing in intensity and in geographic distribution driven by urbanisation, global trade and travel, climate change, and ineffective vector control, with the latter being partly the result of insecticide resistance [3]. Insecticide-based methods are essential to virtually all vector control programmes and, as such, insecticide resistance is a major threat to the control or prevention of all vector-borne diseases. Perhaps most importantly, the control of malaria vectors in Africa has stalled as insecticide resistance may explain recent losses in the effectiveness of insecticide-treated bed nets, which were until now the most effective method for protecting vulnerable communities from malaria infection in Africa [4]. The increase in insecticide resistance among mosquito species is threatening the efficacy of insecticide-based control strategies and behavioural studies are needed to refine the current approaches and develop new interventions [5,6,7,8].

Control approaches for mosquito-transmitted diseases include genetic-based strategies aimed at reducing mosquito densities or replacing wild populations with individual’s refractory to diseases’ infection [4]. This is achieved by releasing mass-produced mosquitoes, usually males, bearing the desired trait(s), aimed at sterilising wild females or spreading the desired genes into the field populations. Mating becomes, therefore, the crucial event for ensuring the success of these strategies, with manipulated males needing to compete with their wild counterparts to inseminate females. Lab-produced males often show a competitive mating disadvantage which can make the mosquito line not suitable for field release [7,8,9]. Therefore, improving the reproductive fitness of lab-produced mosquitoes bearing the desired traits and intended for field release becomes a powerful tool for the success of these genetic approaches. 

Mosquito mating is one of the least understood behaviours of the mosquito life cycle [10], and by gaining more knowledge of this process, it may be possible to identify new opportunities for mosquito control and interventions [4]. Mosquitoes, like many other flying insects, are able to mate in flight. In *Anopheles gambiae* sl., the most important malaria vector worldwide, mating occurs in swarms where 5 to 500 male mosquitoes fly usually above a landmark (a region of high contrast) at dusk [10,11]. Female mosquitoes will fly into the swarm and mate with a particular male, then leave the swarm and usually seek a blood meal. *Aedes aegypti* and *Aedes albopictus*, major arbovirus vectors in tropical and temperate regions, mate in small swarms present in the cooler hours of the day on top of dark landmarks and water containers. Often, males fly around the host waiting for females in host-seeking behaviour and inseminate them while approaching the host for a blood meal [7,8]. Recent studies on mating revolve around the acoustic signals produced in mate selection as a result of the wing beat frequency [4,12]. However, tracking the flight of mosquitoes is still in its infancy, with Butail et al. being one of the first to quantify the interactions within wild mating swarms of male *Anopheles gambiae* [13]. The data gathered through these experiments are used in this paper to further understand and model mosquito behaviour.

There are few studies that have applied machine learning to insect trajectories. One example of a research paper that resembles the work presented here is the automatic identification of phlebotomine sand flies, in which Machraoui et al. attempted to identify sand flies among other insects [14]. An artificial neural network (ANN) was used to classify the two classes of insect flight and obtained performance close to 88% for both accuracy and F1 score when using an optimised set of trajectory features. The approach consisted of ensuring track lengths contained between 14 to 56 positions, with features later extracted that are used in the classification of the trajectories. These track lengths span a large range that may impact the values produced by curvature-based features, as longer tracks have more opportunity for deviations within flight than shorter tracks. There has been little exploration into the classification of mosquito trajectories, which can reveal new insights into mosquito behaviour that may be utilised to reduce the spread of disease.

In this paper, we propose a novel methodology that can classify and identify behavioural differences between male and non-male mosquitoes. The novelty of this approach lies in splitting tracks into track segments and later combining them using a voting method, as well as the use of unique features of flight that have yet to be applied to mosquito trajectories. Furthermore, we use an anomaly approach where we define a one-class method (i.e., we consider a male class against any known or unknown class of non-males) instead of a fixed number of classes and use explainable artificial intelligence (AI) techniques to extract scientific insights into the behavioural differences between these classes. Overall, the methodology standardises the track duration, extracts kinematic and shape-descriptor features, and uses a Support Vector Machine (SVM) classifier alongside a voting method that overall separates the male and non-male classes of mosquito tracks. Using this system, biologically relevant conclusions on the differences between mosquito behaviours of separate classes are obtained, which are supported by data and experts’ knowledge, rather than only expert opinions. The system is tested on 3D tracks generated from natural mating swarms of the malaria vector mosquito *Anopheles gambiae* in the field in West Africa. Through the development of this system, the differences between different male and non-male classes of mosquitoes are quantified and interpreted from a biological perspective. We believe such work will add to the ongoing research into mosquito behaviour.

## 2. Materials and Methods

To classify male and non-male mosquito tracks, we divided the task into four parts. First, raw tracks were split into segments, then features were extracted from these segments, and, finally, a subset of features was selected to be classified by a machine learning model. Together, these steps form a system that has the potential to work on a variety of trajectory-based domains. Throughout the paper, a mosquito track was considered as a three-dimensional track or trajectory, T, that can be described as follows, where $x_{i}$, $y_{i}$, and $z_{i}$ correspond to the ith position within a Cartesian coordinate system, and $t_{i}$ is the time:(1)$$T=\left\{\left(x_{i},y_{i},z_{i},t_{i}\right)\;\;\mathrm{for}\;i=0,\ldots ,N\right\}\;$$

### 2.1. Dataset Description

The dataset used to validate our pipeline was generated by Butail et al. and reported in 2009, 2010, and 2011 [13,15]. Data collection was completed in the village of Donégu ébougou, Mali, where mosquito swarms of *Anopheles gambiae* formed around 20 min after sunset. Three-dimensional mosquito trajectories were generated through the processing of stereoscopic recordings, as described in [13,15]. Phase-locked Hitachi KP-F120CL cameras were used to capture 10-bit images at 25 frames per second at a 1392 × 1040-pixel resolution. The x- and y-axes of the dataset were parallel to the ground, with the *x*-axis parallel to the bisector of the stereo cameras. The *z*-axis was therefore vertical, parallel to the gravitational field. The stereo camera system was set up parallel to east-west, such that the *x*-axis corresponded to east-west. 

In summary, the dataset contains 12 male mosquito collections contained 191 tracks and 10 mosquito couple collections containing 753 tracks (where the male and female mosquitoes were mating while in flight and were tracked as a single entity). These collections were generated from different mosquito swarms where male tracks were recorded and no females were present, and couple tracks were generated from swarms that contained mating events. The couple was tracked a few seconds prior to the detection of a female within the measurement volume and ended when the couple flew out of the field of view. There were also 6 female tracks and 6 corresponding focal-male tracks (males that were pursuing a female and ended up mating) that were tested. The mating pairs were identified through manual detection after recording. Female mosquitoes proved to be more challenging to detect due to their faster movement compared to males, often appearing as a blur in the footage. The majority of the swarm consisted of males, making them more identifiable.

The swarms were almost exclusively monotypic with respect to molecular form, therefore, they only encompassed one species of mosquito. The mosquito data captured were from wild mosquitoes mainly of the Savanna chromosomal form, with few swarms being of the Mopti form. (See [16] for details on these chromosomal forms.) An illustration of the trajectories for the 4 mosquito classes is given in Figure 1, with (a) showing the male swarming behaviour (blue) and the shorter track of a mating couple (red), and (b) showing the focal male (blue) and female (red) prior to becoming a mating couple.

### 2.2. Data Processing

Individual trajectories hold information on both the movement of a mosquito, but also artefacts of the recording process. It was found that different track classes had varying track durations due to the recording process—couples (when mosquitoes mate in flight and are tracked as a single unit) only last for a few seconds whereas male tracks can be minutes long. Similarly, females tend to spend less time within swarms as shortly after entering a swarm, they form a couple and soon leave. To mitigate the effect that this duration imbalance may have on classification, a windowing technique was used to split tracks into equal-sized segments. Given a specified window size and overlap, the algorithm splits tracks to return equally sized (in duration) track segments that correspond to the window size. The overlap corresponds to the length of time that was shared with consecutive track segments, which helps to reduce the loss of information at segment ends. This is named a windowing technique as it is analogous to a window frame moving along the track and taking a snapshot. To obtain a suitable segment length and overlap, these parameters were optimised, as described in Section 2.6.

### 2.3. Feature Extraction

From the track segments, various features of flight were extracted to produce a rich feature set that was used in classification. Various properties of the segments were computed which returned a single metric or multiple values. Where multiple values were returned (such as with velocity which was determined at each time step), a statistical summary was computed for those properties for each track segment. The statistical summary consisted of the mean, median, standard deviation, 1st quartile, 3rd quartile, kurtosis (a measure of the tailedness of a distribution), skewness (a measure of the asymmetry of a distribution), the number of local minima and maxima, and the number of zero crossings. 

#### 2.3.1. Velocity, Acceleration, and Jerk

These kinematic features were extracted using second-order central finite difference methods. The features include absolute values of velocity-based features, signed values of acceleration and jerk, and axial values for each of the x-, y-, and z-axes [17]. Raw values of velocity were not included as a mosquito can only fly forward (hence, in the positive direction).

#### 2.3.2. Angle of Flight

The angle of flight, $\alpha_{i}$, describes the angular change in the path of the mosquito. In other words, the relative change in direction between positions. It is calculated using the dot product of two vectors: (2)$$v_{i}=\left[\begin{array}{c} x_{i}-x_{i-1}\\ y_{i}-y_{i-1}\\ z_{i}-z_{i-1} \end{array}\right]$$
(3)$$v_{i+1}=\left[\begin{array}{c} x_{i+1}-x_{i}\\ y_{i+1}-y_{i}\\ z_{i+1}-z_{i} \end{array}\right]$$
(4)$$\alpha_{i}=\arccos\left(\frac{v_{i}\cdot v_{i+1}}{\sqrt{\left(v_{i}\cdot v_{i}\right)\left(v_{i+1}\cdot v_{i+1}\right)}}\right)$$

#### 2.3.3. Angular Velocity and Angular Acceleration

The angular velocity and angular acceleration were calculated using the angle of flight, $\alpha_{i}$. Four variations for each feature were calculated: the projections to the X-Y, Y-Z, and X-Z planes, as well as a feature encapsulating all planes (3D).

#### 2.3.4. Orthogonal Components of Velocity

These are a group of metrics that describe the tendency of a movement to head in a given direction [18]. The persistence velocity describes the tendency to head tangential to the trajectory. The turning and inclination velocities are the tendencies to head normal to the trajectory and orthogonal to each other. To calculate these metrics, a conversion from Cartesian to spherical coordinates $\left(r,\theta ,\phi \right)$ was computed, as well as the change in angles, θ, and ϕ. Then, a conversion back to Cartesian coordinates produced the final metrics. 

To convert from the Cartesian to the spherical coordinate system, the following equations were used:(5)$$\rho_{i}=\sqrt{{\left(x_{i+1}-x_{i}\right)}^{2}+{\left(y_{i+1}-y_{i}\right)}^{2}+{\left(z_{i+1}-z_{i}\right)}^{2}}$$
(6)$$\theta_{i}=\arctan\left(\frac{y_{i+1}-y_{i}}{x_{i+1}-x_{i}}\right)$$
(7)$$\phi_{i}=\arccos\left(\frac{z_{i+1}-z_{i}}{\sqrt{{\left(x_{i+1}-x_{i}\right)}^{2}+{\left(y_{i+1}-y_{i}\right)}^{2}+{\left(z_{i+1}-z_{i}\right)}^{2}}}\right)$$

The change in θ and ϕ was computed as follows, as well as the calculation for the instantaneous velocity, v:(8)$$\Theta_{i}=\theta_{i+1}-\theta_{i}$$
(9)$$\;\Phi_{i}=\phi_{i+1}-\phi_{i}$$
(10)$$v_{i}=\frac{\rho_{i}}{t_{i+1}-t_{i}}$$

Then, to convert back to the Cartesian coordinate system:(11)$$P_{i}=v_{i}\sin\left(\Phi_{i}\right)\cos\left(\Theta_{i}\right)$$
(12)$$T_{i}=v_{i}\sin\left(\Phi_{i}\right)\sin\left(\Theta_{i}\right)$$
(13)$$I_{i}=v_{i}\cos\left(\Phi_{i}\right)$$
where $P_{i}$, $T_{i}$, and $I_{i}\;$are the persistence, turning, and inclination velocities, respectively.

#### 2.3.5. Straightness

The straightness, also known as tortuosity, of a track is the ratio of the actual distance travelled and the shortest path between the start and end positions [19]. A value close to 1 indicates a straight track where the object travels in a direct path to its destination.
(14)$$S=\frac{\sum_{i=0}^{N}\sqrt{{\left(x_{i+1}-x_{i}\right)}^{2}+{\left(y_{i+1}-y_{i}\right)}^{2}+{\left(z_{i+1}-z_{i}\right)}^{2}}}{\sqrt{{\left(x_{N}-x_{0}\right)}^{2}+{\left(y_{N}-y_{0}\right)}^{2}+{\left(z_{N}-z_{0}\right)}^{2}}}$$

#### 2.3.6. Convex Hull

Generally, the convex hull of a set of points is the smallest possible convex polygon (in 2D) or polyhedron (in 3D) that bounds all of the points of a track [20]. Using this notion, various metrics can be extracted from the shape of the convex hull, including the volume and surface area. Below is an image (Figure 2) illustrating the convex hull in 2D, where the set of points joined by the black line forms the convex hull.

#### 2.3.7. Centroid Distance Function

The centroid distance function calculates the distances of each point in a track to the centre point of the track [21]. It is used as a measure of straightness, where values that are constant describe a smooth and circular motion, while motion that involves sudden changes in direction has a centroid distance function that varies.
(15)$$x_{c}=\frac{1}{N}\overset{N}{\underset{i=0}{\sum }}x_{i}$$
(16)$$y_{c}=\frac{1}{N}\overset{N}{\underset{i=0}{\sum }}y_{i}$$
(17)$$z_{c}=\frac{1}{N}\overset{N}{\underset{i=0}{\sum }}z_{i}$$
(18)$$C_{i}=\sqrt{{\left(x_{i}-x_{c}\right)}^{2}+{\left(y_{i}-y_{c}\right)}^{2}+{\left(z_{i}-z_{c}\right)}^{2}}$$

#### 2.3.8. Curvature

The curvature is another measure that defines the deviation of a trajectory from a straight line [21] but focuses on how much a trajectory bends or turns. The curvature is calculated from the projections on each of the X-Y, Y-Z, and X-Z planes taking the mean and standard deviation for each track.

The rate of change of a variable, ${\dot{a}}_{i}$, and the rate of change of ${\dot{a}}_{i}$, i.e., ${\ddot{a}}_{i}$, can be calculated as:(19)$${\dot{a}}_{i}=\frac{a_{i+1}-a_{i}}{t_{i+1}-t_{i}}$$
(20)$${\ddot{a}}_{i}=\frac{{\dot{a}}_{i+1}-{\dot{a}}_{i}}{t_{i+1}-t_{i}}$$

Thus, the curvature, $k_{i}$, can be calculated as: (21)$$k_{i}=\frac{\dot{x_{i}}\ddot{y_{i}}-\dot{y_{i}}\ddot{x_{i}}}{{\left({\dot{x}}_{i}{}^{2}+{\dot{y}}_{i}{}^{2}\right)}^{\frac{3}{2}}}$$

#### 2.3.9. Curvature Scale Space

The curvature scale space, introduced by Mokhtarian and Mackworth, measures the inflection points in a trajectory at different scales and has proven to be effective at matching and recognising shapes that have been distorted by affine transformations, such as translation, rotation, and scaling [22]. The computation for this feature involves finding the points of inflection of a shape at varying levels of smoothing.

To compute the curvature at varying levels of detail, the trajectory is convolved with a 1D gaussian kernel, $g\left(t,\sigma \right)$, of width σ:(22)$$g\left(t,\sigma \right)=\frac{1}{\sigma \sqrt{2\pi }}e^{\frac{-t^{2}}{2\sigma^{2}}}$$

As the value of σ increases, the corresponding curvature zero-crossing point locations are recorded. The values of the locations of zero-crossings at different levels of smoothing can be plotted to produce a CSS image in the $\left(u,\;\sigma \right)$ plane, where u is the normalised arc length and σ is the width of the Gaussian kernel. The plot displays peaks at the inflection points of a shape and is robust to affine transformations and noise. For the later statistical summary, a 1D signal is generated by taking column maximums, and the mean and standard deviation of this parameter are used in the feature set [23].

#### 2.3.10. Fractal Dimension

The fractal dimension is a metric that can be used to describe the amount of movement in a trajectory by measuring how complex or irregular a trajectory is. To appreciate the fractal dimension, we considered an object’s movement [24]. If its motion is along a straight path, then its fractal dimension is d=1. On the other extreme, we considered the movement to emulate planar Brownian motion. In this scenario, the fractal dimension is d=2, where such paths visit all of the points in a 2D plane. An organism’s movement rarely can be considered to have a fractal dimension d=1 or d=2 (or d=3 in 3D space). Instead, the fractal dimension is typically between these extremes, where values near d=1 characterise more linear movement and values closer to d=2 are more like planar Brownian motion. As such, this provides a measure for the sinuosity or tortuosity of a path.

The following equation was used to compute the fractal dimension of any self-similar fractals:(23)$$d=\frac{\log n}{\log\frac{1}{S}}$$
where n is the number of miniature pieces in the path, S is the scaling factor. and d is the fractal dimension. For a trajectory, the values of  n and S were defined as below.
(24)$$n=\overset{N}{\underset{i=0}{\sum }}\sqrt{{\left(x_{i+1}-x_{i}\right)}^{2}+{\left(y_{i+1}-y_{i}\right)}^{2}+{\left(z_{i+1}-z_{i}\right)}^{2}}$$
(25)$$S=\frac{1}{\sqrt{{\left(\max x-\min x\right)}^{2}+{\left(\max y-\min y\right)}^{2}+{\left(\max z-\min z\right)}^{2}}}$$

### 2.4. Feature Selection and Anomaly Detection Model

Following the feature extraction process, a subset of features was selected. The Mann-Whitney U-test was chosen as it is a non-parametric test and does not require an assumption on the underlying distribution. Using the SciPy Python package [25], this test was used to determine whether there were any statistical difference between the normal class (males) and the abnormal class (non-males). To reduce Type I errors, a family-wide error rate correction using Bonferroni adjustments, and computed using StatsModels [26], was applied with a family-wise error rate (α) of 0.01. The features that reject the null hypothesis at a 0.01/n significance level were selected, where n was the number of features. From this set of features, one out of each group of highly correlated (85%) features was preserved. 

The anomaly detection, also known as novelty detection, attempted to detect whether a new observation was an outlier or not. In our case, an inlier was a sample that was identified as a male. To perform the anomaly detection, a one-class classifier known as the one-class SVM classifier, computed using scikit-learn [27], was chosen. The one-class SVM model attempts to create a decision boundary around the datapoints that it has been trained on, and, in this case, it was trained on the male-only data. The model provided various hyperparameters, including 𝜈 and the SVM kernel type, which were found via hyperparameter tuning. The 𝜈 value was described as “an upper bound on the fraction of training errors and a lower bound of the fraction of support vectors” [27]. The kernel parameter transformed the input data into a higher-dimensional feature space where it can be more easily separated. The choice of kernel allowed for the modelling of nonlinear relationships between the input data, which may not be easily separated by a linear decision boundary. A new observation was classified depending on which side of the decision boundary it lies.

The one-class SVM model was used alongside a voting classifier. Each track segment was given a prediction by the model and the mode of the segments for each track was computed to give a final prediction of the class of the track. If there was an equal number of predictions for each class for a track, it was classified as anomalous (i.e., non-male).

### 2.5. Evaluation and Interpretation

To evaluate the performance, a K-fold cross validation approach was used on the male trials after the validation trials had been removed; 2 trials of male mosquitoes were iteratively removed for the testing set, and the rest of the male data was used to train the model. Overall, the data used to test the model consisted of the 2 trials of male tracks, the remaining couple tracks (after removal of those used for validation), and all of the female and the focal-male tracks. Each iteration is known as a fold, where a different combination of 2 male trials were used in testing and the rest of the male tracks were used for training. Each subclass (male, female, focal-male and couple) was considered separately and together, so that it was possible to identify how each class performed. Prior to the model fitting, the data were standardised via Z-score normalization. The mean and standard deviation were calculated from the male training data and applied to all of the test data. 

To evaluate the performance of the pipeline, various metrics were computed, including accuracy, balanced accuracy, recall score, precision score, F1 score, ROC AUC score, and PR AUC score using scikit-learn [27]. The equations for these metrics can be found in Appendix A. It should be noted that the PR AUC score and ROC AUC score are usually computed using probability scores; however, the One-Class SVM model does not support converting a decision to a probability score. Instead, the signed distance to the hyperplane was used as a proxy after having mapped it to a probability distribution using the logistic function. This was appropriate as large distances indicate more certainty in a prediction.

After the training of the machine learning model, it was possible to interpret how the model made predictions from which we could gain insights regarding the mosquitoes’ flight. For this, SHAP (shapely additive explanations) [28] values were calculated and plots were generated. These values described the contribution that each feature had to the model. The SHAP values were computed using the segment predictions of the best and worst-performing folds, determined by balanced accuracy. From the values, plots can be generated and combined with expert knowledge in order to obtain insights.

### 2.6. Hyperparameter Tuning

A few parameters of the model were tuned. These include the one-class SVM parameter, ν, and the kernel, as well as the window size and overlap length of the windowing technique used to split tracks into segments. The parameters were tuned together in a cross-validated grid search approach to find the combination of parameter values that maximises the balanced accuracy score over the complete test dataset. Tuning was conducted using 3 male trials and 2 couple trials selected in order to give approximately equal numbers of segments in the male and non-male sets, forming the validation dataset. These 5 trials were kept independent of the datasets used for the training and test. The search was conducted based on the parameter ranges given in Table 1. 

## 3. Results

The processing pipeline was evaluated against the tracks generated from mosquito mating swarms. Here, a detailed description of the processing approach and an evaluation of the model is provided. The following section describes the initial interpretation of behaviours of male and non-male mosquitoes using our methodology.

### 3.1. Data Processing

Each mosquito track was split into equal-sized segments of 1.6 s using a windowed approach, where the size of the overlap between segments was 0.8 s. These values were optimised through the grid search approach described above. 

In [13], a trial conducted on 28 August 2010 that contained only male tracks was described to be anomalous due to high wind speeds during the recording. As such, the mosquitoes displayed a rolling motion in the direction of the wind and gained unusually higher velocities. This effect was observed in a scatter plot of the track duration against displacement (Appendix A), where tracks from this trial lay outside a confidence interval of 95% evaluated over all of the trials in terms of temporal and physical track length. Similarly, a boxplot of the velocities in each male trial (Appendix A) illustrated this trial as an anomaly. As such, this trial (ID 5) was removed from the dataset. The results from the methodology using this trial can be found in Appendix A, which demonstrates the poor performance due to the effect of strong weather conditions. The tracks shorter than double the segment length were also removed—so that at least two segments existed for each track. Table 2 contains a summary of the total number of track segments after processing. The ‘before’ and ‘after’ columns denote the change in the number of tracks from the initial dataset to the processed dataset, as described above. Typically, mating couples lasted for a short duration before separating, and thus, after processing, fewer numbers of couple tracks were returned. 

### 3.2. Model Evaluation

The system aimed to classify male and non-male mosquito tracks, and, as such, the model was trained on exclusively male mosquito tracks. The one-class model returned data points that were either within the decision boundary, the normal class: males; or were outliers, the anomaly class: non-male. The model hyperparameters were optimised using a grid search approach as described previously which led the hyperparameter, 𝜈, to be set to 0.20, as well as the kernel to be set to the radial basis function (RBF). 

Overall, 28 folds were computed, the scores for each track segment were combined for each track via the voting method, and the scores from each fold were averaged. The final scores based on the track classification are presented in Table 3, Table 4 and Table 5. The 95% confidence interval is also provided in brackets.

Note that the male-only and focal-male tracks were expected to be classified as male, and the couple and female sets were expected to be classified as non-male. The values in Table 3, Table 4 and Table 5 are the performances with respect to the expected outcome. Figure 3 displays a confusion matrix that illustrates the sum of the predictions for the entire dataset of whole tracks for all computed folds. Figure 4 displays two PR curves for the whole track predictions where male and non-male tracks were the positive class in the respective parts of the figure, including the best- and worst-performing folds. Figure 5 displays a ROC curve for each fold in light blue, with the average highlighted in dark blue. A grey shadow indicates the standard deviation.

Figure 6 and Figure 7 display SHAP plots that identify the contribution that each feature had to the model [28]. Figure 7 reveals that all selected features contributed roughly equally. 

## 4. Discussion

With experiments using mosquito mating swarms, the scores displayed a good performance to discriminate between the male and non-male classes despite the variability in the dataset. The data used to validate the proposed method were generated in the field using stereo-cameras. Such a method introduces noise where environmental factors can influence the mosquito flight, including weather, odours, and animal activity. Such variability would not be observed in tracks generated in a more controlled setting. An inherent limitation from the use of stereo-cameras is that the depth positional values (with respect to the cameras, x-coordinates) are less accurate than values in the other axes. On the other hand, there is a possibility of other mosquito species or other flying insects participating within the swarm without the knowledge of the investigators. The dataset mostly contains male and couple tracks, alongside a few female and focal-male tracks. As the couples themselves contain both a male and female mosquito, it is plausible that couples may display some male-like flight behaviour. A significant difference for a couple is its increased inertia compared to a male mosquito while the combined aerodynamic forces available from the male and the female are also increased. These factors combined result in a dataset that may be hard to classify and hence it is considered that the proposed method returns a strong performance. The highest average accuracy obtained is for the female group at 100% throughout all folds, while it is recognised that there are only 6 tracks of this type. Similarly, the focal-male group obtained an accuracy of 83.3% throughout all folds where there were only 5 tracks available.

From Table 3 and Figure 3, it can be seen that the balanced accuracy and ROC AUC scores are fairly strong where the SVM model is able to distinguish between male and non-male classes. Through the confusion matrix, Figure 3, it is shown that the model can identify true males and true non-males. 

The anomaly classification model adopted is particularly appropriate for the available data where the different track groups (male, female, focal male and couple) exhibit a large contrast in population, as shown in Table 2. The model also gives the flexibility to test several anomalies against the single (male) class. A more balanced dataset would enable binary classification models to be evaluated. However, as within mating swarms, there are fewer female mosquitoes, so a single-class approach was appropriate. 

The methodology described allows for the exploration of acute differences in the flight rather than general swarming behaviours. Mosquito behaviour can easily be described using whole tracks rather than segments, as whole tracks contain an important behavioural difference—the time of flight. It has been shown that using whole track duration and distance travelled features out-perform the current methodology with a balanced accuracy of 80.1% and a ROC AUC of 84.3% (for full results see Appendix A). Naturally, within mating swarms, female mosquitoes and couples are visible for a short time because of their expected biological mating behaviours; in contrast, males fly in orbits generating longer duration tracks. Furthermore, the stereo imaging system has a spatially fixed and finite field-of-view; hence, the intrinsic limitations of the imaging system would also lead to female tracks being shorter than males—an artefact that is unrelated to flight behaviour. This provides additional motivation to examine detailed flight behaviour using metrics based on limited duration segments, in this case 1.6 s, rather than building the best classifier. 

To date, the application of machine learning techniques on mosquito trajectories has received little attention. A similar previous study focused on classifying phlebotomine sand flies [14] and reported high accuracy rates using machine learning models. However, it is important to note that these studies classified distinctly different species as well as a further study on sandflies before and after exposure to a large amount of repellent. In contrast, our research aims to differentiate between subtle differences in the movements of male and non-male mosquitoes within mating swarms, which presents a greater challenge due to the complex nature of swarm behaviour and background noise in field recordings. Despite the lower performance of our model, the results indicate that there are indeed distinguishable differences between these classes, highlighting the potential for further exploration of machine learning methods in analysing mosquito behaviour.

Mention should be made of the large confidence intervals for the male test set. Upon inspection, it was found that a few similar trials, one conducted on 6 October 2011 (named trial 8) and two trials conducted on 7 October 2011 (named trials 9 and 11), performed strangely during the model evaluation using cross validation. These three trials were the only trials in the test set that were of the Mopti form rather than the Savannah form, as well as the only trials where the swarm formed over a bundle of wood rather than bare ground. When these tracks were included in the test set, generally, those male test results were quite poor. However, when they were included in the training set, the male test performance improved. The couple performance was generally consistent through all folds. The top 5 performing male test folds mostly contained these trials (8, 9 and 11) in the training set and trial 2 in the test set. As trials 8, 9, and 11 are of a different form and swarming over a different marker, some variability in their flight is expected. The variability in particular features in these trials can significantly impact the performance of a model. Including such variability in the training set helps the model fit well across the range of values and helps classify all males. However, in some cases, the model may overfit to this variability resulting in highly accurate classifications of males but leaving less room for non-males to be classified as anomalies. Furthermore, trial two′s features are likely to have much less variability and, thus, fit well within the decision boundary created by the large variation in these other trials.

Although there may be slight variations in the behaviour of the mosquitoes in different trials, incorporating them into the model provides a more generalised representation of mosquito swarming behaviour. Nevertheless, from Appendix A, there are very few tracks/segments in these trials, and, as such, their removal would not significantly affect the model’s performance (as seen in Appendix A). Despite this, it was ultimately decided to retain these trials in the model as they may still contribute to the overall understanding of mosquito behaviour. Through the inclusion of these trials, we can observe how different mosquito forms and environmental factors influence the model’s performance, highlighting the importance of accounting for such factors. 

The PR curves for the male set display unusual results. The curves look as if they start at (0,0); however, they do start at (0,1) but drop towards the origin instantly at the first few threshold. Usually, the points with highest probability are classified correctly (true positive), so the precision should be around one. However, in this case, there is a very small number of points with the highest probabilities that are actually false positives and lead to a precision of zero. This is observed in some of the other folds too, where the worst folds are those that contain the trials 8, 9, and 11 in the test set. The performance in these folds may be caused by the different mosquito form and swarming marker, but the other folds may display this performance due to the variability generated from experiments in the field. 

From the SHAP plots, it is possible to extract behavioural insights. Generally, there is noise in these plots, possibly as a result of using the data gathered in the field. Due to this noise and some anomalous feature values, the colour scale representing the magnitude of the feature values might be slightly skewed. Nevertheless, to ensure the robustness of our interpretations of the model, the SHAP summary plots were used in conjunction with scatter plots to assess the feature value distributions (an example scatter plot can be found at Appendix A which contains mostly red data points; however, this is due to the influence of extreme negative values—hence, examining the distribution of colours in the SHAP summary plots alone can be misleading). The summary and scatter plots reveal that the positive SHAP values display only slight positive values, while the negative SHAP values are largely negative. This is likely due to the proximity of feature values for the normal instances in the dataset to the decision boundary defined by the one-class SVM model. As a result, features with positive SHAP values contribute very little to the classification decision, while anomalous features that are further away from the decision boundary contribute much more and are assigned very negative SHAP values. The bar plots show that all features contribute roughly equally, which indicates that the model uses the complex interactions between the selected features rather than a contribution from a select few. The ranked lists of mean absolute SHAP values (bar plots) for the best and worst folds show significant differences, but there are similarities in the feature value contributions (summary plots). The fold that performed the best included trials zero and two, which displayed velocities that were comparable to many other trials and well within their velocity distributions, as shown in Appendix A. On the other hand, the worst-performing fold included trials seven and eight, where trial eight was previously noted to contain a different mosquito form and swarming marker. Trial seven does contain mosquitoes of the Savannah form; however, the swarming marker is also different (grass patch rather than a bundle of wood). As such, much of the discussion draws insights from the SHAP plots generated from the best-performing fold.

Upon initial examination, it is evident that extreme values across most features contribute towards a non-male classification, particularly in kinematic features. In the case of radial acceleration and jerk-based features (radial velocity was not selected from our feature selection process) that capture all 3D components (unlike the axial-based features), larger or extreme features values tend to contribute more toward a non-male classification. This is observed in the first and third quartiles of radial acceleration, where the largest and the smallest values tend to lead to a non-male classification, for each feature, respectively. The same trend is displayed in the first and third quartiles of radial jerk, as shown in Figure 8, which overall shows that extreme values tend to lead to a non-male classification. The pattern observed in the best-performing fold can be explained by the fact that non-male mosquitoes, which include mosquito mating couples and females, are generally larger and require more energy to maintain flight than a single swarming male, resulting in more erratic flight and more changes in velocity. As mating couples contain two bodies, issues may arise when they attempt to coordinate flight which is, mechanically, a much more complex process. Some kinematic features are also described using the number of zero crossings and the number of local minima and maxima, but they are difficult to interpret as they contain a lot of noise. Nevertheless, these features rank lower and contain much smaller absolute SHAP values in comparison to those describing the first and third quartiles, and thus contribute less to the model.

The axial kinematic features, including the x, y, and z components of velocity and acceleration, show very similar trends, where larger or extreme values contribute to a non-male classification. Non-male mosquitoes tend to display larger values of acceleration and velocity in the x and y components, which continues to follow the previous trend and may be explained by non-males struggling to maintain stability in flight while trying to stay together. The z components (height above ground) of velocity and acceleration show that extreme values tend to lead to a non-male classification. This can be attributed to the fact that male mosquitoes fly with relative stability when swarming in circular motions with little change in their vertical direction, whereas couples may experience chaotic flight during mating, leading to extreme values in both velocity and acceleration in the z-axis.

Some of the highest contributing features were orthogonal components of velocity that describe the mosquito’s tendency to move in a specific direction. Inclination velocity measures the degree to which a mosquito is inclined to move up or down as it moves forward. Notably, non-males exhibit extreme values in inclination velocity, again, describing males as being relatively stable in the z-axis. While persistence velocity was also selected through the feature selection, its contributions were much lower and more challenging to interpret.

The SHAP plots indicate that angle-based features do vary between males and non-males in mating swarms, possibly reflecting the distinct roles each group plays in the mating process. Specifically, non-males exhibit much lower angles of flight values than males, or, in other words, lower changes in direction. However, the non-males have much larger values in angular acceleration and angular velocity. Combined, this describes non-male trajectories as rapidly changing and unstable, but with smaller angle changes in comparison to males. This is consistent with the chaotic nature of mosquito mating couples, as well as reflects the large angle changes observed in males whilst flying in circular motions within a mating swarm.

Meanwhile, shape descriptors provide insight into the shape of mosquito tracks. Although most shape descriptor features have little impact on model predictions, their contributions offer expected outcomes. For instance, straightness and fractal dimension features reveal that non-male mosquitoes tend to have extreme values, indicating both high and low amounts of curvature in flight, which is in line with expectations. Similarly, male trajectories also exhibit smaller ranges of curvature than non-males, as described by the SHAP plots where the standard deviation of curvature in all projections are lower. 

Overall, the SHAP values validate our prior expectations regarding flight within mosquito mating swarms that male mosquitoes have much more stable flights than non-male mosquitoes. Due to the nature of mating swarms, male mosquitoes fly relatively stable while waiting for a female. On the other hand, female mosquitoes will search for a mosquito mating swarm, and once in a swarm, will couple with a male fairly quickly. During the mating process, mosquitoes have been observed to engage in brief and intense periods of copulation while in flight, during which their flight becomes erratic. This behaviour is likely due to their need to maintain a close proximity to one another in order to successfully mate. Nevertheless, the signal within the SHAP plots is noisy due to the variability of natural mosquito populations and experimental conditions.

It was noted that the most contributing features are velocity- and acceleration-based, and few shape descriptors were selected in the feature selection process. Similarly, those few shape descriptors that were chosen had little contribution to the model classification. It was expected that some shape descriptors may display some variability that would enable the model to separate the classes; nevertheless, the results show that on the level of individual track segments, shape descriptors struggle to separate sexes. Shape descriptors may be expected to have a greater influence for larger segment durations. Whilst the optimum segment duration was found to be 1.6 s, in hyperparameter tuning, segment durations were explored up to 10 s but were not selected due to the poorer performance of the model, in terms of balanced accuracy. Further exploration of the dataset would be needed at larger segment durations to understand the role of trajectory shape descriptors as a means of distinguishing sex.

## 5. Conclusions

In this paper, a methodology was proposed for an anomaly detection system that could identify male mosquito tracks in mosquito mating swarms. The track segmentation method and the feature extraction processes are, to our knowledge, novel in their application to mosquito trajectories. Following those steps, a feature selection stage utilising the Mann-Whitney U-test was used to select features with the highest variability. A cross validation approach was used to evaluate the model’s performance, where the track segments were combined with a voting method. Even though 3D tracks were used in the case study considered, this pipeline with minimal modifications could be applied to 2D tracks (albeit with a possible degradation in performance due to loss of one axis of information). The model performance was affected by individual trials, which under field conditions were strongly influenced by severe weather conditions and variations in the environment.

This pipeline could also be used for various trajectory problems where there is a high imbalance of tracks or for comparing multiple classes to a reference class. Using the mosquito swarming dataset provided by Butail et al. [13], the pipeline was proven to work even with noisy data generated in the field with a balanced accuracy of 64.5% and a ROC AUC score of 68.4%. The results presented in this paper showed the efficiency of the method to identify male mosquito trajectories from non-males. Such a system is useful to further the understanding of mating mosquitoes and, thereby, give insights into the mechanisms that may limit mosquito breeding and disease transmission.

## Figures and Tables

**Figure 1 biology-12-00496-f001:**
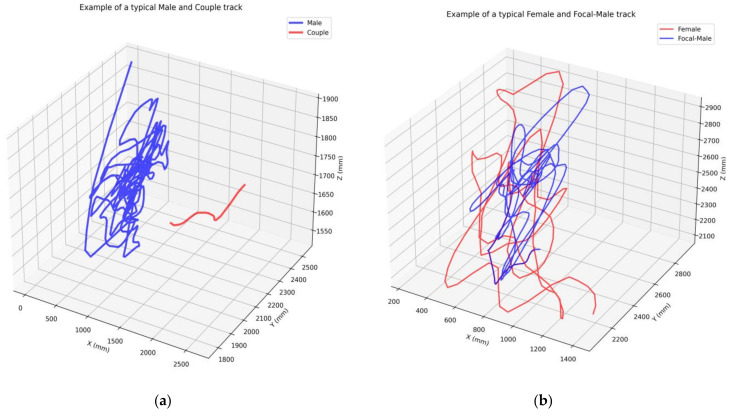
Example of typical tracks. (**a**) Example of a typical single male mosquito and a mating couple track. The red and blue lines display the mating couple and single male paths, respectively. (**b**) Example of a mosquito mating sequence. The red and blue lines display the female and focal-male paths, respectively. At the end of this sequence, they form a mating couple.

**Figure 2 biology-12-00496-f002:**
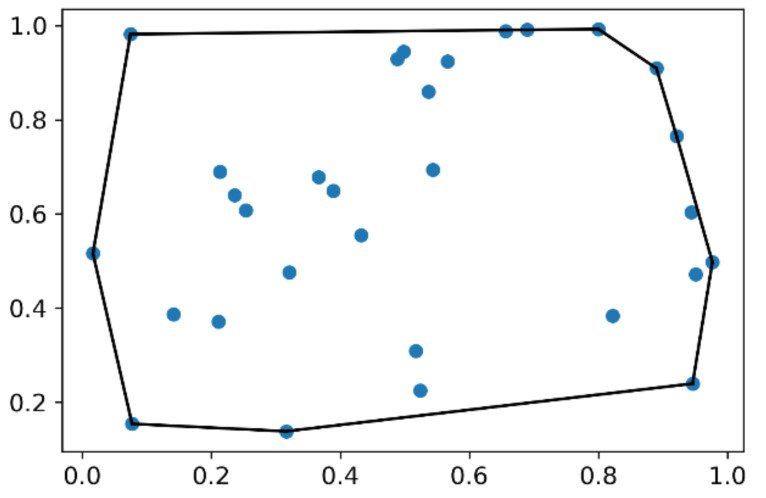
Example of convex hull which is the set of points joined by the black line.

**Figure 3 biology-12-00496-f003:**
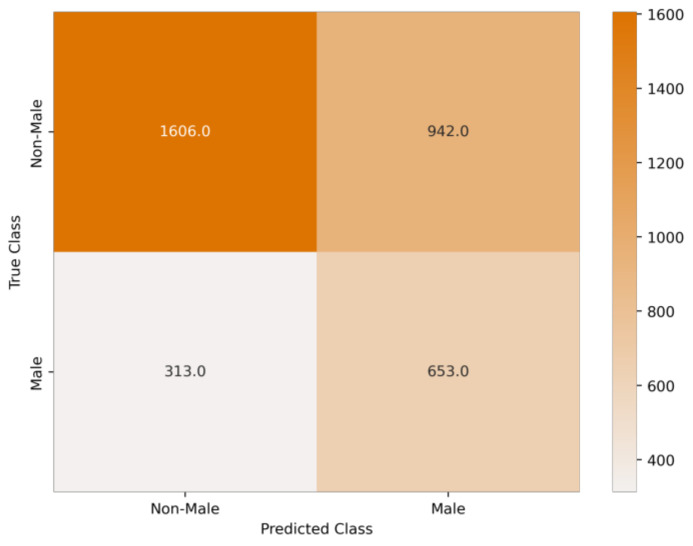
Confusion matrix for the predictions of whole tracks of all folds.

**Figure 4 biology-12-00496-f004:**
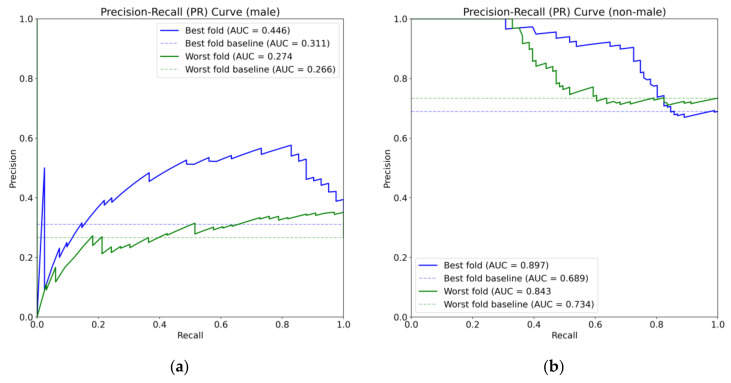
PR curve for predictions of the tracks. (**a**) PR curve where males are positive class. (**b**) PR curve where non-males are positive class.

**Figure 5 biology-12-00496-f005:**
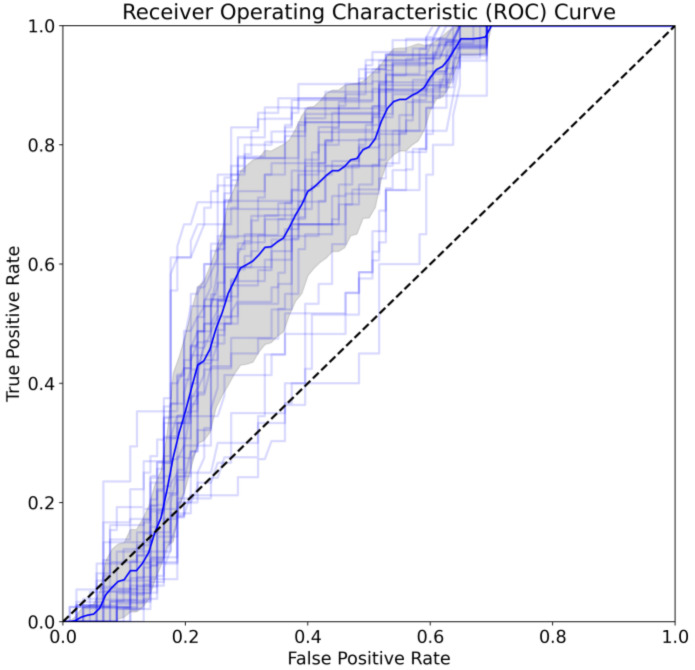
ROC curve for predictions of the tracks displaying the performance at various classification thresholds. All folds are included in light blue with the average indicated in the dark blue and the standard deviation in the grey shadow.

**Figure 6 biology-12-00496-f006:**
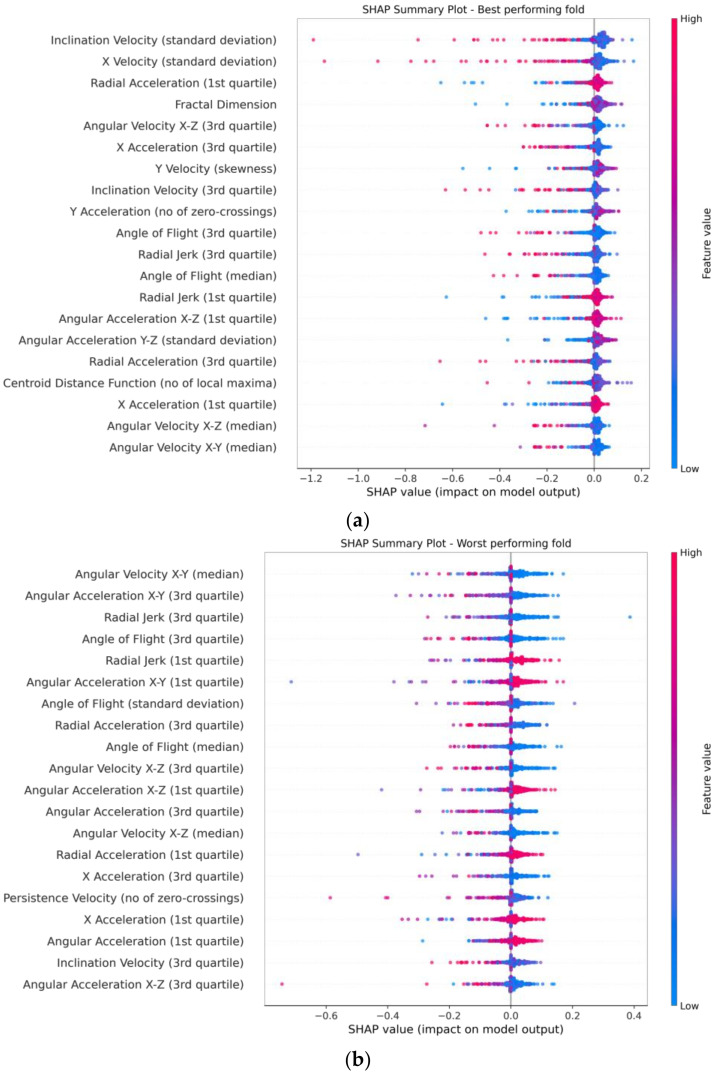
SHAP summary plot displaying the contributions of each feature to the model. The features are ordered by absolute feature contribution and the top twenty are displayed. A full list can be found in Appendix A. Each dot represents a segment, with its colour displaying its feature value. Positive SHAP values contribute towards the male class, while negative values contribute to the non-male class. (**a**) Summary plot of the best-performing fold. (**b**) Summary plot of the worst-performing fold.

**Figure 7 biology-12-00496-f007:**
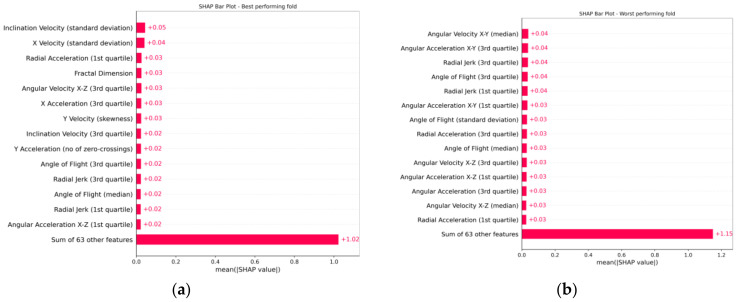
SHAP bar plot of the contributions of each feature to the model. The features are ordered by absolute feature contribution and the top fifteen are displayed. A full list can be found in Appendix A. (**a**) Bar plot of the best-performing fold. (**b**) Bar plot of the worst-performing fold.

**Figure 8 biology-12-00496-f008:**
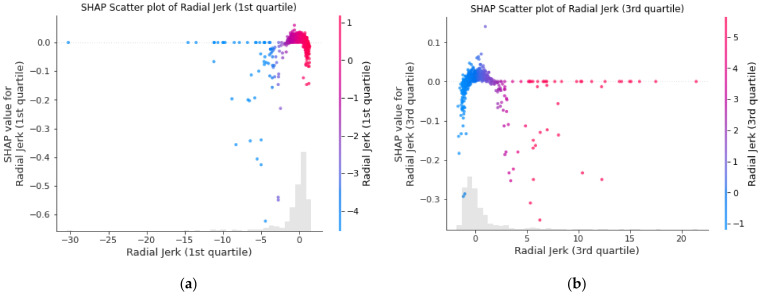
SHAP scatter plot of the (**a**) 1st and (**b**) 3rd quartiles of radial jerk displaying the distribution of contributions of each feature value.

**Table 1 biology-12-00496-t001:** Hyperparameter tuning parameter ranges.

Hyperparameter	Range	Step Size
*𝜈*	0–1	0.01
Kernel	linear; radial basis function; polynomial; sigmoid	-
Window Size	1–10 s	0.2 s
Overlap Length	0.2–5 s	0.2 s

**Table 2 biology-12-00496-t002:** Number of tracks and segments following initial processing.

Dataset	Total Number of Tracks before	Total Number of Tracks after	Number of Segments
Male Tracks	191	161	3012
Couple Tracks	753	119	1059
Female Tracks	6	5	27
Focal-Male Tracks	6	6	69

**Table 3 biology-12-00496-t003:** Classification performance of each class.

Dataset	Accuracy
Train set (Male only)	0.733 (0.674–0.787)
Test set (Male only)	0.609 (0.238–0.867)
Test set (Couple only)	0.616 (0.581–0.670)
Test set (Female only)	1.000 (1.000–1.000)
Test set (Focal Male)	0.833 (0.833–0.833)

**Table 4 biology-12-00496-t004:** Classification performance of complete test set.

Dataset	Balanced Accuracy	ROC AUC Score
Test set (complete set)	0.647 (0.499–0.746)	0.694 (0.577–0.759)

**Table 5 biology-12-00496-t005:** Classification performance of normal and anomaly classes of complete test set.

	PR AUC Score	Precision	Recall	F1 Score
Male	0.357 (0.210–0.466)	0.396 (0.225–0.555)	0.656 (0.366–0.862)	0.489 (0.283–0.660)
Non-Male	0.885 (0.847–0.926)	0.841 (0.734–0.926)	0.637 (0.604–0.688)	0.724 (0.682–0.755)
Overall	0.621 (0.551–0.676)	0.619 (0.499–0.716)	0.647 (0.499–0.746)	0.607 (0.490–0.702)

## Data Availability

The data presented in this study are available on request from Butail et al. The code used in this paper has been deposited to and made publicly available from the authors’ GitHub repository, https://github.com/yasserqureshi1/finding-a-husband (accessed on 8 February 2023).

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
