# Peer review of "Finding a Husband: Using Explainable AI to Define Male Mosquito Flight Differences"

_biology, 2023, doi:10.3390/biology12040496_

Round 1
Reviewer 1 Report
This study presents a machine learning pipeline that classifies male versus non-male mosquitoes using flying trajectories collected in the field. The authors addessed the problem of imbalanced duration by segmenting each trajectory into overlapped windows, extracting relevant features, training one-class SVM, and combining the predictions in a winner-take-all manner. Overall, the pipeline is effective in classifying sex and has the potential for providing further insights into mosquito behavior.
This topic is of interest to a general audience, the manuscript is well-written, and the methods are adequately explained. However, the data presentation can be improved and some additional analyses should be done. I recommend accepting the paper after the following issues are addressed.
-
One novelty of this study is track segmentation, yet some key data were not provided. For example, the distbution of the number of segments per track and the distribution of their votes (likely corresponding to the classification confidence).
-
Similarly, the authors should show some raw distributions of extracted features, at least for the leading features in SHAP plots.
-
Table 2: Why were so many coupled tracks removed?
-
Figure 2 (and also Figure 5): It is surprising to see that the recall of the male test set was so low (674 / (674 + 306)), given that the classifier was trained using the male data. In the paragraph starting from Line 496, the authors discussed the possible reason of the large confidence intervals for the male test set, which I found not very satisfying. If trials 8, 9, and 11 were found problematic, the authors should retrain the model with those trials excluded.
-
Figure 5: The dots are difficult to distinguish and appear to be inconsistent with the main text. For instance, the authors note on Line 541 that females with lower absolute jerk contribute to the negative SHAP, but most of the blue dots seem to be on the right side. It is also confusing that the authors then state on Line 542 that females are expected to have larger jerk, which contradicts the prior sentence.
Minor points:
- Line 400: wrong table name
- Line 505 and 512: should it be trial 1 or trial 11?
Reviewer 3 Report
It is a relevant work that brings a lot of information that can be of great value for the control of mosquito vectors. Artificial intelligence is a path that will give us many answers and support for the development of new approaches through the behavioral study of the mosquito. There are minor revisions needed throughout the manuscript.

Reviewer 4 Report
-
Brief summary:
-
The authors present a novel machine learning model for insect flight classification. In this study, the methodology is applied to Anopheles gambiae mating swarms captured in the field in west Africa (Mali) using stereo-cameras. The authors use their pipeline to distinguish male from non-male (i.e., female and mating pairs) mosquito flight by analyzing 3D flight tracks by segment and then combining segment outcomes to classify the entire track. This innovative technique allows flight components to be broken down into their constituent behavioral features and scored based on their respective contributions to the model.
-
-
Broad comments:
-
The manuscript is well written and clear. The results are promising and show potential for further optimization. Further, this study presents preliminary data indicating that this pipeline could be used to distinguish not only sexes, but also species and forms. Although this method was “only” 75.8% accurate in identifying male vs non-male flight trajectories, and thus lower than comparable methods in other insects (e.g., 88% accuracy in distinguishing sand flies from other insects in Machraoui et al., 2018), this outcome is actually fairly impressive when one considers the complexity of male swarms and the noise involved in field recordings. Nor would 100% accuracy (such as observed here for the “female only” test set) be required in every application, for instance in differentiating between species or between lab and field strains. Since one would not expect to see totally different flight patterns in these cases anyways, but rather more or less overlap in similarities, one could set thresholds far below 100% distinct that would be sufficient for spotting significant differences in flight patterns. Given the merits of this paper’s findings, I recommend this article for publication in Biology.
-
-
Specific comments:
-
Lines 72-73: The authors state that “Mosquitoes are one of the few insects that are able to mate in flight.” Can the authors verify this claim? On the face of it, this statement does not appear to convey how common swarm mating is among flying insects.
-
Round 2
Reviewer 1 Report
The manuscript has been significantly enhanced and my concerns have been effectively addressed. I have no further comments.
Reviewer 2 Report
I am satisfied with the authors comments and edits to the manuscript. I don't have further comments.